

# Prevalence of HPV genotypes and assessment of their clinical relevance in laryngeal squamous cell carcinoma in a northeastern state of Brazil—a retrospective study

Charlles Brito[1], Rachel D. Cossetti[2,3], Diego Agra de Souza[3], Marcos Catanha[3], Pablo de Matos Monteiro[4] and Flavia Castello Branco Vidal[1,5]

[1] Programa de Pós-Graduação em Saúde do Adulto, Universidade Federal do Maranhão, São Luís, Maranhão, Brazil
[2] Departamento de Medicina I, Universidade Federal do Maranhão, São Luís, Maranhão, Brazil
[3] Departamento de Patologia, Instituto Maranhense de Oncologia Aldenora Belo, São Luís, Maranhão, Brazil
[4] Departamento de Farmácia, Universidade Federal do Maranhão, São Luís, Maranhão, Brazil
[5] Departamento de Morfologia, Universidade Federal do Maranhão, São Luís, Maranhão, Brazil

Corresponding author
Flavia Castello Branco Vidal,
flavia.vidal@ufma.br

## ABSTRACT

**Background**. A high prevalence and incidence of head and neck tumors make Brazil the country with the third-highest number of cases of these malignant neoplasms. The main risk factors are smoking and alcohol consumption; however, cases related to the human papillomavirus (HPV) have tripled in number, demonstrating a changing disease profile. Studies have reported the prevalence of HPV in laryngeal squamous cell carcinoma (LSCC) to vary between 8% and 83%. The role of HPV as an important causative factor in LSCC remains unclear.

**Methods**. This retrospective study included 82 patients with LSCC diagnosed between 2014 and 2019 at two oncology hospitals in São Luís, Brazil. Sociodemographic and clinical data, and the histopathologic characteristics of the tumors, were collected directly from medical records. Genetic material was extracted from paraffin-embedded samples using nested polymerase chain reaction (PCR) and automated sequencing for HPV detection and genotyping. The results by social and clinicopathologic variables were then compared using the chi-squared test and multivariate analysis.

**Results**. Sociodemographic analysesdemonstrated that most patients were men (87.8%), brown-skinned (75.6%), and resident in the state capital (53.7%). They generally had a poor education status (53.7%), having only an elementary school education (completed/incomplete), and 51.2% were self-employed in occupations such as farming or fishing. Smoking and alcohol consumption habits were observed in approximately half the patients. With respect to clinical characteristics, 39% of patients exhibited T1/T2 staging, 51.2% had no distant metastasis, and 30.5% had lymph node invasion. HPV DNA was detected in half the samples (50%), with the high oncogenic type 16 being the most prevalent. There was no significant relationship observed between the economic, educational, occupational with the HPV LSCC in the presented data, although multivariate analysis demonstrated that HPV DNA was more likely to be present in T3–T4 tumors ($p = 0.002$).

## INTRODUCTION

A high incidence and prevalence of head and neck squamous cell carcinoma make Brazil the country with the third highest number of cases of this malignant neoplasm (*Johnson et al., 2020*). The mortality for this disease is high depending on the stage of the lesion at diagnosis, the disease being the sixth leading cause of death in the world (*Cohen et al., 2019*; *Johnson et al., 2020*).

Head and neck squamous cell carcinoma constitutes a diverse group of cancers with different behaviors and prognoses, necessitating different treatment approaches. The disease can occur at various sites, such as the oral cavity, oropharynx, nasopharynx, hypopharynx, larynx, paranasal sinuses, and salivary glands (*Cohen et al., 2019*). Exposure to carcinogens, including tobacco and alcohol, is the dominant causative factor. In the past decade, the human papillomavirus (HPV) has been found to play a role in oropharyngeal carcinogenesis, and HPV involvement in other head and neck sites has become more prominent (*Amit et al., 2016*).

The role of HPV in squamous cell carcinoma of the oropharynx is well established; however, its role in laryngeal squamous cell carcinoma (LSCC) remains unclear. The reported prevalence of HPV in LSCC varies greatly between studies, ranging from 8% to 83% (*Zhang et al., 2016*; *Erkul et al., 2017*; *Tong et al., 2018*; *Vazquez-Guillen et al., 2018*; *Yang et al., 2019*). The type of HPV most often isolated in laryngeal tumors is type 16, generally followed by type 18 (*Kariche et al., 2018*; *Dogantemur et al., 2020*).

LSCC is the second most common head and neck squamous cell carcinomas (*Vazquez-Guillen et al., 2018*). It is highly metastatic, with a 5-year survival rate that does not exceed 50% for stage IV tumors (*Kariche et al., 2018*). It occurs principally between the fifth and seventh decade of life and predominantly affects men (*National Cancer Institute of Brazil, 2020*). In Brazil, the annual age-standardized incidence rate per 100,000 men is 5.33, compared with 4.18 in South America and 3.59 worldwide (*ICO, 2016*). The mortality incidence is also higher in Brazil, at 4.45 compared with 3.21 in South America and 2.17 worldwide (*ICO, 2016*)

As in head and neck cancer, HPV is an important prognostic factor in LSCC, considered to possibly influence prognosis and treatment decisions (*Huang & O'Sullivan, 2017*). There are few Brazilian studies on laryngeal cancer associated with HPV infection, and none have included patients from the state of Maranhão, which is in a very poor region of Brazil. Our aim in the present study was therefore to conduct a retrospective analysis of the prevalence of HPV and its genotypes as factors in the development of laryngeal cancer in the affected population in the northeastern region of Brazil.

## MATERIALS & METHODS

### Enrollment and ethics approval

In this retrospective study, we analyzed formalin-fixed, paraffin-embedded laryngeal squamous cell carcinoma specimens collected at two public oncology referral hospitals in Maranh ao, a state in northeastern Brazil. The study included 82 samples from patients diagnosed with laryngeal carcinoma between January 2014 and December 2019.

Patient information and the histopathologic characteristics of the tumors were obtained from medical records. Because the samples consisted of paraffinized tumors, written informed consent was not required to be obtained from the patients. Patient identities were not disclosed in the study, which was approved by the Ethics in Research Committee of the Federal University of Maranhao (register number 3.023.486).

### Inclusion criteria

Patients were included when paraffin blocks and histology slides of laryngeal carcinoma, whether from biopsy or surgical treatment at any follow-up stage, were available in the archives of the pathology services.

### Exclusion criteria

Patients were excluded if their medical records and/or paraffin tumor blocks were not found or were not sufficient for laboratory analysis in the study.

### DNA extraction and HPV analysis

The samples were reviewed by a pathologist, and blocks representative of tumor (containing more than 50% of the total area of the fragment) were selected.

For DNA extraction, 15 sections of approximately 5 μm thickness containing biologic material were used. The sections were stored in 2.0 mL tubes at 4 °C until the DNA extraction step.

Genomic DNA was extracted from the samples using the ReliaPrep FFPE gDNA Miniprep kit from Promega (Promega, WI, EUA), according to the extraction protocol suggested by the manufacturer. To assess the concentration and quality of the biologic material, the samples were quantified using a Nanodrop Lite spectrophotometer. Concentrations were reported in nanograms per microliter, and the DNA quality was verified by reading the 260/280 nm wavelength ratio. When the ratios were in the range 1.7–1.9, the material was considered pure.

Nested PCR reactions were performed using a Veriti 96-Well Thermal Cycler (Applied Biosystems, Thermo Scientific, California, USA), with primers PGMY09 and PGMY11 for the first round and primers GP + 5 and GP + 6 for the second round (*Vidal et al., 2016*).

The samples were purified using the GenElute PCR Clean-Up (Merck KGaA, Darmstadt, Germany) Kit according to the manufacturer's instructions and were subsequently quantified. Automated sequencing was performed by ACTGene Análises Moleculares company which currently provides DNA sequencing services, using the AB 3500 platform (Applied Biosystems, Thermo Scientific, Waltham, MA, USA). Samples were diluted in 6 μL of a solution containing 30–60 ng purified amplification product, 5 picomol of primer, and double-distilled water.

To confirm and identify the HPV type, the nucleotide sequences of sequenced samples were compared and submitted to the World Nucleotide Database (GenBank) using the BLAST program.

## Statistical analysis

Categorical variables are presented as percentages, and the chi-squared test was used for comparisons between groups. Statistical significance was set at $p < 0.05$. After the univariate analysis, significant variables ($p < 0.05$) were selected for binary logistic regression analysis to assess their relationship with HPV infection.

## RESULTS

The analysis showed that most patients were men (87.8%), brown-skinned (75.6%), and from the capital of the state (53.7%). Furthermore, 53.7% had only an elementary school education (completed/incomplete), and 51.2% were self-employed in occupations such as farming or fishing. Laryngeal cancer staging data showed that 29.3% had stage N0 disease; 39%, stage T1/T2; and 51.2%, stage M0. Finally, 51.2% were nonsmokers, and 57.3% were nonalcoholic (Table 1).

HPV analyses by nested PCR demonstrated that half the samples were positive for the virus. The patients were therefore divided into two groups: HPV+ and HPV–.

In univariate chi-squared analysis between the two groups, only residential location, T stage, and smoking status were statistically significant ($p < 0.05$, Table 1). A multivariate binary logistic regression analysis was then performed to verify the relationships between those variables and the presence of HPV.

In multivariate analysis, residential location was not significant in terms of its relationship with HPV infection, and smoking was not an explanatory variable for HPV infection. In contrast, a T3/T4 disease stage decreased the chances of the patient belonging to the HPV- group by 0.234 (CI 0.066 to 0.837, $p < 0.05$; Table 2) and increased the chances of the patient belonging to the HPV+ group.

Figure 1 shows the HPV types found in HPV+ laryngeal cancer samples. The high oncogenic risk types 16, 45, and 33 were observed in most patients (68.3%). Type 6, a low-risk HPV, was found in 1 patient. In 12 samples, determination of the HPV type by the automated sequencing technique was not possible. HPV 16 was the most prevalent type in this cohort ($n = 22$, 53.65%).

## DISCUSSION

According to recent publications, the mean age of patients with laryngeal cancer is 65 years, with a higher proportion of men than of women (*Kariche et al., 2018*; *Tong et al., 2018*; *Koroulakis & Agarwal, 2022*). Our patients were predominantly men, and they were slightly younger, with a mean age of 62 or 63 years (HPV+ and HPV- groups).

Analyzing geographic distribution, patients with HPV+ laryngeal cancer came mainly from the capital, while patients with HPV–laryngeal cancer came primarily from the interior of the state. The difference was statistically significant by the chi-squared test, but that significance was lost when binary logistic regression was applied. Differences between

**Table 1  Sociodemographic and clinical data for 82 patients with laryngeal squamous cell carcinoma by HPV status.**

| | | Total (N = 82) | | Human papillomavirus | | | | P-value[a] |
|---|---|---|---|---|---|---|---|---|
| | | | | Positive (N = 41) | | Negative (N = 41) | | |
| | | N | % | N | % | N | % | |
| Age (Years) | Mean (±) | | | 62.24 ± 8.7 | | 63.26 ± 9.2 | | 0.5 |
| Sex | Female | 10 | 12.2 | 6 | 14.6 | 4 | 9.8 | 0.5 |
| | Male | 72 | 87.8 | 35 | 85.4 | 37 | 90.2 | |
| Ethnic group | White | 9 | 11 | 4 | 9.8 | 5 | 12.2 | |
| | Black | 8 | 9.8 | 3 | 7.3 | 5 | 12.2 | 0.7 |
| | Brown | 62 | 75.6 | 32 | 78 | 30 | 73.2 | |
| | Other | 3 | 3.7 | 2 | 4.9 | 1 | 2.4 | |
| Origin | Interior | 38 | 46.3 | 15 | 36.6 | 23 | 56.1 | 0.05 |
| | Capital | 44 | 53.7 | 26 | 63.4 | 18 | 43.9 | |
| Schooling | Illiterate | 9 | 11 | 6 | 14.6 | 3 | 7.3 | |
| | Elementary school | 45 | 54.9 | 23 | 56.1 | 22 | 53.7 | 0.4 |
| | High school | 28 | 34.1 | 12 | 29.3 | 16 | 39 | |
| Ocupation | Retired | 32 | 39 | 17 | 41.5 | 15 | 36.6 | |
| | Autonomous | 42 | 51.2 | 23 | 56.1 | 19 | 46.3 | 0.8 |
| | Other | 8 | 9.8 | 1 | 2.4 | 17 | 17.1 | |
| | No | 24 | 29.3 | 13 | 31.7 | 11 | 26.8 | 0.4 |
| | Yes | 25 | 30.48 | 11 | 44 | 14 | 56 | |
| Lymph node invasion | | | | 3 | 7.3 | 8 | 19.5 | |
| | | | | 1 | 2.4 | 2 | 4.9 | |
| | | | | 1 | 2.4 | 0 | 0 | |
| | Not evaluated | 33 | 40.24 | 17 | 51.5 | 16 | 48.5 | |
| | T1/T2 | 32 | 39 | 16 | 39 | 16 | 39.0 | 0.05 |
| T stage | T3/T4 | 20 | 24.4 | 14 | 34.1 | 6 | 14.6 | |
| | Notevaluated | 30 | 36.6 | 11 | 26.8 | 19 | 46.3 | |
| M stage Distant metastasis | No | 42 | 51.2 | 20 | 48.8 | 22 | 53.7 | 0.5 |
| | Yes | 1 | 1.2 | 1 | 2.4 | 0 | 0 | |
| | Not evaluated | 39 | 47.6 | 20 | 51.3 | 19 | 48.7 | |
| Smoking | Yes | 40 | 48.8 | 16 | 39 | 24 | 58.5 | 0.05 |
| | No | 42 | 51.2 | 25 | 61 | 17 | 41.5 | |
| Alcohol consumption | Yes | 47 | 57.3 | 25 | 61 | 22 | 53.7 | 0.5 |
| | No | 35 | 42.7 | 16 | 39 | 19 | 46.3 | |

**Notes.**
Data are presented as means with standard deviation, frequencies, and proportions.
[a]By chi-squared test with significance accepted at 95%; *$p < 0.05$.

patients depending on the geographic regions in which they live are possibly attributable to different social and sexual habits (*Hoffmann & Quabius, 2021*).

Smoking is an important risk factor in the pathogenesis of laryngeal tumors (*Zhang et al., 2016*; *Erkul et al., 2017*; *Tong et al., 2018*; *Koroulakis & Agarwal, 2022*). Generally, non-smoking patients are mostly HPV+, and smokers are mostly HPV−(*Xu et al., 2014*;

**Table 2  Multivariate analysis of the associations between HPV infection and sociodemographic characteristics of patients with laryngeal squamous cell carcinoma.**

| | Human papillomavirus | | |
|---|---|---|---|
| | *Odds ratio –( 95% CI)* | *Wald Statistic* | *P-value* |
| *Origin* | | | |
| **Interior** | 0.425 (0.166–1.091) | 3.165 | 0.075 |
| **Capital** | Ref | | |
| **T staging** | | | |
| T1/T2 | 0.588 (0.193–1.607) | 1.170 | 0.279 |
| T3/T4 | 0.234 (0.066–0.827) | 5.083 | **0.002** |
| Not evaluated | Ref | | |
| **Smoking** | | | |
| **Yes** | 0.484 (0.191–1.230) | 2.325 | 0.127 |
| **No** | Ref | | |

Notes.
Data were evaluated by binary logistic regression analysis. Statistical significance was set at $p < 0.05$.
OR, odds ratio; CI, confidence interval.

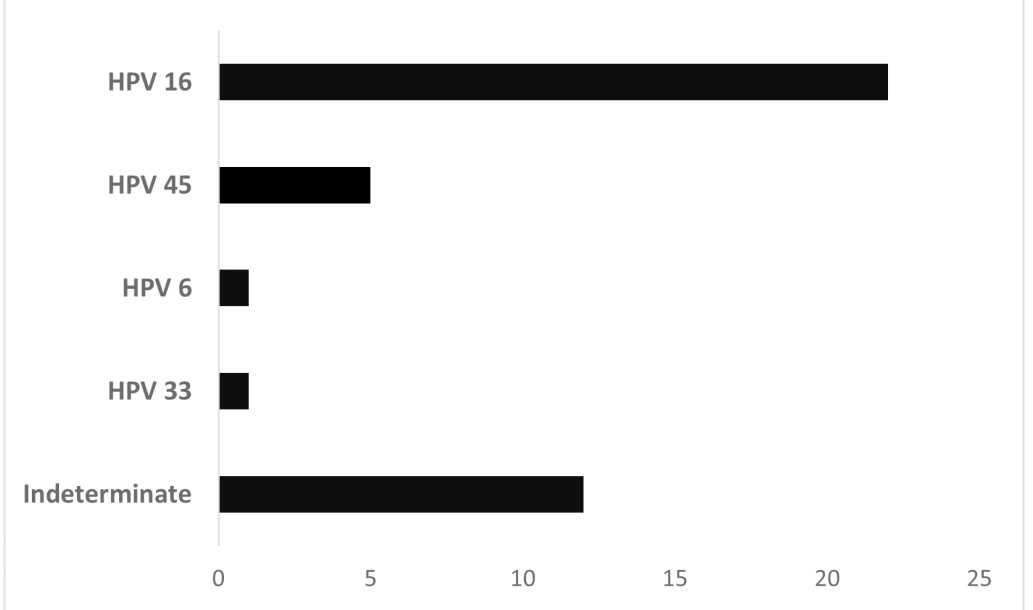

**Figure 1  Distribution of laryngeal cancer patients ($n = 41$) stratified by HPV type.**

*Hoffmann & Quabius, 2021*). A similar distribution was observed in our study population. In the HPV+ group, non-smokers predominated, whereas in the HPV–group, most patients were smokers ($p < 0.05$). However, this variable's statistical significance was lost when binary logistic regression was applied.

In other studies, no significant relationship between smoking and the presence of HPV has been found; both variables should therefore be regarded as independent factors in the oncogenic process of the larynx (*Sánchez Barrueco et al., 2017*; *Dogantemur et al., 2020*).

Alcoholism is also described as a risk factor for this type of tumor; however its use was not related to HPV infection which was observed by others (*Erkul et al., 2017*; *Dogantemur et al., 2020*).

Unfortunately, TNM classification data were not available for all patients. N0 and M0 staging were prevalent among the analyzed tumors. No statistical association with HPV infection was observed, as in other studies (*Xu et al., 2014*; *Erkul et al., 2017*; *Sánchez Barrueco et al., 2017*; *Dogantemur et al., 2020*).

T1–T2 staging was prevalent, but a higher index of T3–T4 tumors was observed in the HPV+ group compared with the HPV–group. By applying binary logistic regression, we observed that patients staged as T3–T4 had a greater chance of having HPV DNA. Similar results were observed in a Turkish population (*Erkul et al., 2017*). Duray and co-authors (*2011*) found that stage IV LSCC cases were shown to have higher HPV positivity compared to stage I and II cases. A Greek study found a significant association between HPV DNA positivity and poorly-differentiated LSCC (*Laskaris et al., 2014*).

In fact, it is currently uncertain whether HPV positivity in LSCC could be used as a prognostic factor. There is strong support in the literature for the association between HPV-positive carcinomas and better prognoses, especially in oropharyngeal carcinomas, but for LSCC the results are still controversial. Generally, author found no difference at all or a slight but nonsignificant improvement in terms of overall or disease-free survival for HPV-positive cases.

Epidemiology studies reported that HPV infection rates in laryngeal cancer range from 8% to 83% (*Zhang et al., 2016*; *Erkul et al., 2017*; *Tong et al., 2018*; *Vazquez-Guillen et al., 2018*; *Yang et al., 2019*). These differences in findings could be attributable to a variety of factors, such as differences in population geographic distribution, ethnic group (different cultures), type of biologic material analyzed (fresh or formalin-fixed), and sensitivity and specificity of the test methods used (*Gama et al., 2016*).

In our study, half the samples contained HPV DNA. For HPV detection, we used formalin-fixed samples and nested PCR with PGMY09/11 and GP5+/6+ primers (*Gravitt et al., 2000*). The sensitivity of this method and its ability to amplify and detect more than 25 of the HPV genotypes, have led to it being considered a "gold standard" for HPV detection (*Abreu et al., 2012*; *Li et al., 2013*; *Hoffmann & Quabius, 2021*).

*Vazquez-Guillen et al. (2018)* reported a 47.7% HPV DNA prevalence in their 195 formalin-fixed LSCC specimens from a Mexican population. They used a molecular method based on reverse hybridization (*Vazquez-Guillen et al., 2018*). *Tong et al. (2018)* analyzed 211 Chinese patients with LSCC. They also used the nested PCR method to detect HPV DNA and found a slightly higher prevalence than ours, 62.6% (*Tong et al., 2018*).

Regarding geographical distribution, systematic reviews and meta-analysis study demonstrated that HPV prevalence is significantly higher in South America, compared to Asia, North America and Europe (*Kreimer et al., 2005*; *Li et al., 2013*). Two Turkish studies revealed HPV DNA prevalence of 12.2% and 26.02% (*Erkul et al., 2017*; *Dogantemur et al., 2020*), similar to a Spanish study in which HPV DNA was detected in 22.76% of the samples (*Sánchez Barrueco et al., 2017*). *Hernandez et al. (2014)* observed an HPV DNA prevalence of 21% in an American population.

A North American study demonstrated that a poor education status and a low income level were related to the presence of HPV DNA in head and neck tumors (*Shewale et al., 2021*). HPV infection is known to be more prevalent in poor and developing countries, where sanitary conditions and access to health care are neither efficient nor sufficient (*de Martel et al., 2020*; *Kombe Kombe et al., 2021*). The Brazilian city in which our study was carried out is in a region that has the second worst human development index in Brazil according to official data (*IBGE-Instituto Brasileiro de Geografia e Estatística, 2010*). Those circumstances might explain the high prevalence of HPV DNA in our samples.

Of the total HPV+ cases, 68.3% had a high risk HPV genotype like those found by *Vazquez-Guillen et al. (2018)* (72%). HPV 16 was the genotype most frequently identified in our study, which is consistent with other studies (*Hernandez et al., 2014*; *Erkul et al., 2017*; *Sánchez Barrueco et al., 2017*; *Tong et al., 2018*; *Vazquez-Guillen et al., 2018*). HPV 18 is the second most common genotype in laryngeal cancers (*Xu et al., 2014*; *Sánchez Barrueco et al., 2017*), although it was not detected in our patients.

HPV 6, a low-oncogenic genotype, was observed in one patient in our study. Other researchers have also found HPV 6 in laryngeal tumor samples (*Castellsagué et al., 2016*; *Kariche et al., 2018*). HPV 6 is highly prevalent in laryngeal papillomatosis, and a small fraction of those cases transform into malignancies (1–4%) (*Carvalho et al., 2021*). No medical history was available for our patient with HPV 6, a 58-year-old man, to allow us to suggest that such a shift had occurred.

Prophylactic HPV vaccination could lead to a worldwide decline in HPV-driven head and neck squamous cell carcinoma (*Hoffmann & Quabius, 2021*). In Brazil, the government has offered a tetravalent vaccine for genotypes 6, 11, 16, and 18 (*Wendland et al., 2021*). Our results show that the tetravalent vaccine covers most cases, but the nonavalent vaccine might be more efficient, because it also includes HPV 31, 33, 45, 52, and 58. Genotypes 33 and 45 were observed in our laryngeal cancer population.

According to the 8th edition of the *TNM Classification for Head and Neck Cancer,* HPV status is an important prognostic factor that can influence treatment decisions (*Huang & O'Sullivan, 2017*). The highest precision in determining HPV status is therefore of utmost importance. Whenever possible, PCR-based methods, still referred to as the gold standard,'' should be used.

A strength of our study is that it is the first to include a northeastern Brazilian population, whose socioeconomic status is significantly lower than that of the better known southeastern regions of Brazil, including S ao Paulo and Rio de Janeiro states (*IBGE-Instituto Brasileiro de Geografia e Estatística, 2010*).

Overall, the sample size was small to run multivariable models, and this is a limitation of our study. Many patients were excluded from the study because the blocks containing the tumor samples were not found and/or the data from the medical records were insufficient. Only recently have the medical records of patients become electronic and easy to find. Formerly, they were kept manually, with poor filling and difficult accessibility, especially for patients who were deceased.

HPV infection may play an important role in the initiation and progression of laryngeal cancer, but the subject still needs clarification, given that HPV infection rates vary greatly between studies.

## CONCLUSIONS

Our findings demonstrate an important prevalence of HPV in a sample of laryngeal epidermoid tumors in a population in northeastern Brazil, with the highly oncogenic type 16 being the most prevalent HPV genotype. In addition, patients with HPV+ LSCC had a higher index of T3–T4 tumors. Our study is the first in Brazil to include only laryngeal tumors in a northeastern population. These data are important for improving our knowledge about the role and consequences of HPV infection in laryngeal tumors. Limitations include clarification of the effect of HPV on prognosis in LSCC, given that some important data were absent from the medical records. Further studies for the evaluation of molecular markers such as p16 and p53 expression are needed to clarify the protein modifications that HPV may cause in laryngeal SCC.

## ACKNOWLEDGEMENTS

We thank the Instituto Maranhense de Oncologia Aldenora Belo and Hospital Geral Tarquínio Lopes Filho for providing access to paraffin blocks, histology slides, and clinical and anatomopathologic data for patients with LSCC, and samples for the research.

### Funding

This study was supported by grants from Fundação de Amparo à Pesquisa e ao Desenvolvimento Científico e Tecnológico do Maranhão (FAPEMA) UNIVERSAL-00634/18. The funders had no role in study design, data collection and analysis, decision to publish, or preparation of the manuscript.

### Grant Disclosures

The following grant information was disclosed by the authors:
Fundação de Amparo à Pesquisa e ao Desenvolvimento Científico e Tecnológico do Maranhão (FAPEMA): UNIVERSAL-00634/18.

### Competing Interests

The authors declare there are no competing interests.

### Author Contributions

- Charlles Brito performed the experiments, analyzed the data, prepared figures and/or tables, authored or reviewed drafts of the article, and approved the final draft.
- Rachel D. Cossetti conceived and designed the experiments, authored or reviewed drafts of the article, and approved the final draft.

- Diego Agra de Souza analyzed the data, authored or reviewed drafts of the article, and approved the final draft.
- Marcos Catanha analyzed the data, authored or reviewed drafts of the article, and approved the final draft.
- Pablo de Matos Monteiro performed the experiments, prepared figures and/or tables, and approved the final draft.
- Flavia Castello Branco Vidal conceived and designed the experiments, analyzed the data, prepared figures and/or tables, authored or reviewed drafts of the article, and approved the final draft.

## Human Ethics

The following information was supplied relating to ethical approvals (i.e., approving body and any reference numbers):

The study was approved by the Ethics in Research Committee of the Federal University of Maranhão (register number 3.023.486).

## DNA Deposition

The following information was supplied regarding the deposition of DNA sequences:

The sequences are available at GenBank:

11L: MG850215.1; 14L: MG850244.1; 18L: MN542782.1; 2L: AF548835.1; 23L: MK716218.1; 24L: MG849618.1; 30L: MG849618.1; 31L: MK387724.1; 32L: EF140818.1; 33L: MG849618.1; 35L: MG850334.1; 36L: DQ422750.1; 37L: HE798674.1; 38L: KU707481.1; 39L: MG849618.1; 40L: MK387724.1; 42L: KY595153.1; 43L: HE798668.1; 44L: DQ218252.1; 48L: LC155251.1; 50L: MG849618.1; 51L: MG849618.1; 53L: MG849618.1; 62L: DQ422750.1; 63L: MG850720.1; 64L: MK387724.1; 65L: HE798674.1; 66L: MK387724.1; 69L: KJ571159.1; 71L: JN617898.1; 72L: LC456609.1; 73L: MG848363.1; 75L: MG847721.1; 76L: MG850288.1; 81L: MG848784.1; 82L: LC155238.1; 83L: MK716218.1; 88L: LC155240.1; 104L: MG848784.1.

## Data Availability

The raw data is available in the Supplementary File.

## Supplemental Information

Supplemental information for this article can be found online at http://dx.doi.org/10.7717/peerj.13684#supplemental-information.

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
