# Peer review of "Prevalence of HPV genotypes and assessment of their clinical relevance in laryngeal squamous cell carcinoma in a northeastern state of Brazil—a retrospective study"

_PeerJ, doi:10.7717/peerj.13684_

## Round 0.1 · original submission · Major Revisions

Please address the concerns of both reviewers and amend the manuscript accordingly.

·

Basic reporting

Overall well-written.

Experimental design

How many HPV types were analyzed, and which ones?

The sample selected in the study is from a referral hospital. How representative is it of a community? Maybe only the advanced cancer are referred to the hospital, and hence the estimation reported in the study is biased given the association of HPV-associated laryngeal cancer with prognosis.

Overall, the sample size is small to run multivariable models and should be acknowledged as a limitation.

Given the small numbers instead of N, M, T stage-wise analysis, why didn't the authors analyze as distant metastasis yes/no, lymph involvement yes/no? Moreover, it's incorrect to choose no information as a reference category for the T stage analysis.

Was there any difference in histologic subtypes by HPV status?

Validity of the findings

The conclusion statement regarding the role of HPV as a risk factor in laryngeal carcinoma can not be derived from this study as there is no control group, and it's not a prospective study.

Additional comments

The discussion is too long and wordy.

Reviewer 2 ·

Basic reporting

The manuscript entitled “Prevalence of HPV genotypes and assessment of their clinical relevance in laryngeal squamous cell carcinoma in a northeastern state of Brazil” by Santo de Aquina et. al, is reporting prevalence of HPV in laryngeal squamous cell carcinoma (LSCC) in a region specific manner. The prevalence of HPV in LSCC has been reported previously elsewhere as well. The only novelty in this work is that it is pertaining to a specific geographic location, northeastern state of Brazil, where there has not been any apparent reporting previously. In addition, the study could prove helpful in deciding the vaccination strategies against specific HPV genotypes prevalent in the region. However, I have some concerns/questions about the manuscript as detailed below:

-In general, the data set presented here is small (n=82) especially when distributing across different stages of LSCC, the observations must be taken with caution. There is too much of extrapolation based on the previously published data rather than the results presented in the manuscript.

-The results section of the manuscript is too brief and seems giving more emphasis on the numbers which have already been mentioned in the Table 1 and are mostly not statistically significant, with p-values >0.05. The statistically significant parameters which are only 3 in the Table 1 should have been emphasized more or maybe shown as figure and the rest of Table 1 should have been either mentioned briefly and or added as supplementary.

-In the Discussion section, authors are applying the observations from other regions to discuss the results from northeastern state of Brazil where most of the parameters studied elsewhere could not be satisfied statistically.

-The important observation of relationship between T3/T4 disease stage and HPV+ incidence should have been explained further with validations from other similar studies.

-In the conclusion of the paper, line 266 to 268: “patients with HPV+ LSCC were generally found to be nonsmokers, to reside in the state capital, and to have a higher index of T3–T4 tumors”, the authors must specifically mention that there was no significant relationship observed in the first two cases (nonsmokers and residence) in their study. Also, make it clear in the abstract as well especially that there was no significant relationship observed between the economic, educational, occupational with the HPV LSCC in the presented data.

-The authors mentioned they could not detect HPV18 genotype (line 232 to 234), the second most common genotype in laryngeal cancers, in their patients. It is intriguing to know the frequency of HPV18 genotype reported in the country or in the adjacent regions?

Experimental design

no comment

Validity of the findings

no comment

---

## Round 0.2 · accepted · Accept

All concerns were adequately addressed and the manuscript was revised accordingly. Therefore, the amended version is acceptable now.